# Natural Products Targeting PI3K/AKT in Myocardial Ischemic Reperfusion Injury: A Scoping Review

**DOI:** 10.3390/ph16050739

**Published:** 2023-05-12

**Authors:** Syarifah Aisyah Syed Abd Halim, Norhashima Abd Rashid, Choy Ker Woon, Nahdia Afiifah Abdul Jalil

**Affiliations:** 1Department of Anatomy, Faculty of Medicine, Universiti Kebangsaan Malaysia, Cheras 56000, Kuala Lumpur, Malaysia; syarifahaisyah@ukm.edu.my; 2Department of Biomedical Science, Faculty of Applied Science, Lincoln University College, Petaling Jaya 47301, Selangor, Malaysia; norhashima@lincoln.edu.my; 3Department of Anatomy, Faculty of Medicine, Universiti Teknologi MARA, Sungai Buloh 47000, Selangor, Malaysia

**Keywords:** medicinal plants, PI3K/AKT, myocardial ischemic reperfusion injury, phytochemicals, cardiovascular disease

## Abstract

This scoping review aimed to summarize the effects of natural products targeting phosphoinositide-3-kinases/serine/threonine kinase (PI3K/AKT) in myocardial ischemia-reperfusion injury (MIRI). The review details various types of natural compounds such as gypenoside (GP), gypenoside XVII (GP-17), geniposide, berberine, dihydroquercetin (DHQ), and tilianin which identified to reduce MIRI in vitro and in vivo by regulating the PI3K/AKT signaling pathway. In this study, 14 research publications that met the inclusion criteria and exclusion criteria were shortlisted. Following the intervention, we discovered that natural products effectively improved cardiac functions through regulation of antioxidant status, down-regulation of Bax, and up-regulation of Bcl-2 and caspases cleavage. Furthermore, although comparing outcomes can be challenging due to the heterogeneity in the study model, the results we assembled here were consistent, giving us confidence in the intervention’s efficacy. We also discussed if MIRI is associated with multiple pathological condition such as oxidative stress, ERS, mitochondrial injury, inflammation, and apoptosis. This brief review provides evidence to support the huge potential of natural products used in the treatment of MIRI due to their various biological activities and drug-like properties.

## 1. Introduction

Coronary heart disease is the third leading cause of global mortality which associated with 17.8 million of deaths yearly [1]. Coronary heart disease occurs most commonly in patients who present with an acute ST-segment elevation myocardial infarction (STEMI). Infarction deprives the heart from necessary oxygen and nutrients to pump blood effectively to the body system [2]. Therefore, rapid reperfusion is much needed to resuscitate the dying area, which involves angioplasty, primary percutaneous coronary intervention, and thrombolytic therapy. Ironically, the perfusion process can be harmful by intensifying the tissue injury known as myocardial ischemia-reperfusion injury (MIRI) [3]. Ischemia-reperfusion injury refers to the situation where the restoration of the blood flow to the particular tissues that were previously deprived of oxygen paradoxically worsens their function and leads to cellular death. Although the re-establishment of blood flow is necessary to save the ischemic tissue, the reperfusion process could lead to additional damage, endangering the survival and function of the organ. Ischemic-reperfusion injury is not only limited to the myocardium but also affects other organs such as the brain, kidney, gut, and skeletal muscle, depending on the impaired blood vessel [4,5,6]. The damage includes fulminant migration of inflammatory cells, oedema and apoptotic processes, an extension of the infarct area and a dysfunctional microcirculation [4,7].

Natural products (NP) derived from plants could be an alternative source of promising drugs to mitigate the harmful aftereffects of MIRI. NP offers various medicinal values and health benefits [8,9,10,11,12], including to the heart attributed to anti-apoptotic, antioxidant, and anti-inflammatory properties [13,14,15,16,17]. Furthermore, accumulating studies suggest bioactive compounds and extracts of NPs improved the viability of cardiomyocytes in post-MIRI, ameliorated endothelial function, and modulated the inflammatory and stress process [16,17]. Thus, numerous research focused on the bioactive compounds of NPs to reduce MIRI.

The phosphoinositide-3-kinases/serine/threonine kinase (PI3K/AKT) is an intracellular signaling pathway involved in the regulation of cellular metabolism, survival, and apoptosis [18]. Many studies supported the role of PI3K/AKT to inhibit apoptosis by improving cellular viability [13,14,19]. Phosphorylation of PI3K begins when ligands bind to the receptor tyrosine kinase. Consequently, phosphorylated PI3K generates a second messenger, phosphatidylinositol (3,4,5)-triphosphate, which in turn activates downstream signaling cascades by mediating through phosphorylation of AKT [20]. In cell survival/apoptosis, PI3K/AKT inhibits the activity of pro-apoptotic proteins by downregulating the expression of the Bcl-2 family, Bim protein via downstream PI3K/AKT/FOXO3a pathway, thus enhancing cell survival [17]. PI3K/AKT also regulates apoptosis via GSK3 by phosphorylating GSK3 isoforms on the N-terminal regulatory site [20].

Accordingly, the damages related to MIRI are reduced when the PI3K/AKT pathway is regulated, implying its significance in cardioprotection [13,14,15]. Activation of PI3K/Akt pathway reduced the apoptosis in the cardiomyocytes and mitochondrial oxidative damage, leading to better survival of the cardiac function [21].Therefore, the study aims to systematically review the existing literature regarding NPs that effectively hinder the MIRI in MIRI models via PI3K/AKT signaling pathway, which may provide insight for investigators and future research.

## 2. Results

### 2.1. Literature Search

A total of 178 studies were found from the database search. After removing the duplicates, 176 articles were shortlisted. After title-abstract screening, 123 articles were excluded because the articles were not relevant to the topic. The reasons for the irrelevance are non-MIRI models (diabetic cardiomyopathy, myocardial fibrosis, cardiotoxicity, and myocardial infarction, high glucose-induced injury in cardiomyocytes, inorganic mercury-induced cardiac injury), other signaling pathways such as PI3-kinase/glycogen synthase kinase 3β/β-catenin pathway, NFkB signaling, IGF-1R/Nrf2 signaling, and the PI3k/Akt signaling pathway were not measured. By full-text screening of the remaining article 53 articles, a total of 39 articles were eliminated based on the inclusion/exclusion criteria; study models used were not MIRI experiment models (*n* = 36), combination of herbs in the treatment (*n* = 2), and no availability of full text (*n* = 1). Thus, a total of 14 articles from January 2016 to January 2021 were selected in this scoping review. A flowchart depicting the search process and study selection is presented in Preferred Reporting Items for Systematic Reviews and Meta-Analyses (PRISMA) for Scoping Review (Figure 1). The characteristics of the selected studies were summarized in Table 1.

### 2.2. MIRI Models Used

Several experimental models were used in the selected review to mimic ischemic-reperfusion injury either using in vivo, in vitro, and ex vivo methods (Table 2). Four studies used in vivo methods only [15,22,23,24], five studies used in vitro methods [13,25,26,27,28], and another five studies used a combined experimental design [14,29,30,31,32]. A study by Yu et al. 2016 compared the effect of pretreatment of gypenoside in both in vivo and in vitro experiments. The results for both experiments showed gypenoside to have a strong capacity to reduce cell apoptotic rates modulated by upregulation of phosphorylated Akt. The results of protein expression of targeted molecules were comparable and similar between these experimental designs.

In the in vivo method, the left anterior descending coronary arteries (LAD) of the animals were ligated to induce ischemia for some period. Subsequently, ligation was released to mimic reperfusion. Even though the research adopted a similar in vivo design, the timings for ischemia and reperfusion were different. For example, Yu et al., 2016, ligated and occluded LAD for 45 min followed by 3 h of reperfusion, Jiang et al., 2016, occluded the LAD for 30 min followed by 4 h of reperfusion, Zeng et al., 2018, introduced ischemia for 45 min, followed by 4 h of reperfusion, Zeng et al., 2018, and Zhao et al., 2018, ligated the LAD for 45 min mimicking ischemia, followed by 12 h of reperfusion and Wu et al., 2016, induced ischemia by ligation of LAD for 30 min, followed by 24 h of reperfusion.

Additionally, IR can also be induced with prolonged cold ischemia (18 h) and followed by 8 h of reperfusion in rat. For the in vitro model, hypoxic-reoxygenation (H/R) injury was established by culturing H9c2 cell in glucose-free DMEM medium placed in the anaerobic glove box or hypoxic chamber containing high percentage of nitrogen gas to mimic hypoxia. Then, the medium of the cells was changed with glucose containing medium and placed in the regular incubator with atmosphere of 95% air. Using similar H/R injury method in H9c2 cells, Wu et al., 2013, induced 8 h of hypoxia followed by 4 h of reoxygenation, Shen et al., 2018, induced 10 h of ischemia followed by 2 h of reperfusion, and Liao et al., 2016, induced 6 h of hypoxia and 12 h of reoxygenation.

In addition, MIRI can be induced ex vivo using Langendorff perfusion model. There were 3 studies using this model. A study by Mei-ping Wu et al., 2016, stopped the perfusion on the isolated heart for 30 min to induce ischemia and then reperfusion for another 30 min, Zun Peng Shu et al., 2019, found the perfusion on the isolated heart was stopped for 45 min to induce cardiac ischemia and then reperfusion occurred for another 60 min, while Yingli Yu et al., 2019, introduced global ischemia for 40 min and reperfusion for 60 min.

### 2.3. Source of Plant

Five studies mentioned the part of the plant, method of extraction, and the solvents used [13,22,23,26,28]. Bark of Cortex Dictamni was aqueously extracted and *Dracocephalum moldavica* was extracted in ethanol [13,23]. HPLC was performed to analyze the chemical constituents in TFDM. The dry powder of *Dracocephalum moldavica* was further extracted in aqueous ethanol and purified to obtain tilianin, which was then identified by HPLC [22]. Whereas the bark of Myrica rubra was extracted with methanol and the plant of Bauhinia championii was extracted with ethanol and further partitioned with ethyl acetate [26,28]. Another five studies purchased processed plant to be tested with the experimental design, whereas another four studies did not mention the source of the natural products [15,30,31,32] (Table 1). The details of plants selected in the articles, including the doses and duration used were presented in Table 3.

### 2.4. Effects of Natural Products in MIRI via PI3K/AKT Signaling Pathway

#### 2.4.1. Gypenoside

Gypenoside is the main component of *Gynostemma pentaphyllum* (Thunb.) Makino [33,34], and exist mainly as dammarane type-triterpene glycosides [35]. In an oxygen–glucose deprivation/reperfusion (OGD/R) cell and MIRI rat models, gypenoside treatment ameliorated impairments in cardiac morphology at a dose of 50 mg/kg, 100 mg/kg, and 200 mg/kg [29]. In the MIRI group, myocardial tissues showed up-regulation of Bax and cleaved caspase-3 and down-regulation of Bcl-2, which were reversed by gypenoside. The administration of gypenoside significantly reduced the apoptotic rates in both rat and cell models and effectively improved the cardiac function as determined by hemodynamic parameters, lactate dehydrogenase levels (LDH), and creatine kinase (CK). Western blotting showed that these effects of gypenoside against MIRI were achieved via inhibition of endoplasmic reticulum stress (ERS) and apoptotic pathway, which subsequently lead to PI3K/AKT pathway activation. Pre-treatment of gypenoside in rats increased the reduction in p-AKTs and p-GSK3β from ischemic-reperfusion (I/R) injury and reduced the ERS markers, namely GRP78, CHOP, cleaved-caspase 12, p-PERK, p-eIF2α, and ATF4 upregulated by I/R injury [29].

#### 2.4.2. Gypenoside XVII

Gypenoside XVII (GP17) is an antioxidant phytoestrogen of the gypenosides group has a similar structure as estradiol [14,36]. GP17-treated-MIRI-H92c cells improved cell viability, reduced LDH release and caspase-3 activity. GP17 at a dose of 1.25–100 µM ameliorated cardiac contractility dysfunction, oxidative damage, and myocardial apoptosis, leading to mitochondrial injury. Mitochondrial injury was associated with mitochondrial permeability transition pore (mPTP) opening, ROS accumulation, and decreased calcium concentration. Mitochondrial apoptosis caused the accumulation of cytochrome-C in the cytosol and increased Bax expression, which was reversed by GP17, thereby suppressing MIRI-induced mitochondrial injury and apoptosis. PERK/eIF2a- and IRE1-related pathway was found to mediate ERS-cardiomyocytes apoptosis. GP17 relieved the ERS and apoptosis by reducing ERS response; GRP78, ERS sensors; PERK and IRE1 levels, as well as pro-apoptotic proteins; p-JNK, CHOP, Bad, Bax, and caspase-12. Preconditioning with PI3K inhibitor and P38/MAPK inhibitor greatly impaired the favorable effects of GP-17 on ERS, mitochondrial injury and apoptosis, suggesting the involvement of PI3K/AKT signaling pathway in the cardioprotective role of GP17 [14].

#### 2.4.3. Geniposide

In a hypoxia/reoxygenation (H/R) of H9c2 cells, pre-treatment with geniposide, an extract from *Gardenia jasminoides* J. Ellis inhibited cardiomyocytes apoptosis, thus improving cell viability and decreased LDH levels [25]. Mitochondrial dysfunction induced by MIRI was reversed by geniposide at a dose of 2.5–320 µM via inhibition of oxidative stress, namely ROS/RNS and MDA, elevating the level T-SOD, improvement of mitochondrial morphology, attenuation of mitochondrial calcium overload, and blunting depolarization of the mitochondrial membrane. Geniposide enhanced the Bcl-2/Bax ratio and p-AKTserine473. In addition, geniposide attenuated the cleaved caspase-3 and cytochrome-c. The protective effects of geniposide were reversed by glucagon-like peptide-1 receptor antagonist exendin-(9-39) (GLP-1R) and the PI3K inhibitor LY294002, which further confirms that geniposide improved H/R-induced myocardial apoptosis by reversing mitochondrial dysfunction through GLP-1R and PI3K/AKT signaling activation [25].

#### 2.4.4. Berberine

Berberine is an alkaloid derivative, isolated from Coptis chinensis Franch [37] and *Berberis vulgaris* L. [38]. In the MIRI mouse model, berberine improves blood lipid levels, blood pressure, and myocardial function (9). Berberine also reduced inflammatory responses via the nuclear factor (NF)-κB signaling pathway. Berberine increased the expression of PI3K/AKT in myocardial cells. Berberine reduced apoptotic protease-activating factor-1 (apaf-1), caspase-3 and caspase-9 expression while up-regulation of Bcl2-associated agonist of cell death, Bcl-2-like protein 1, and cellular tumor antigen p53. In conclusion, berberine suppressed the apoptosis of myocardial cells via the PI3K/AKT signaling pathway at a dose of 10 mg/kg which was administered to rats for 30 days [30]. In a separate study, 25 mg/kg, 50 mg/kg, and 100 mg/kg of berberine were administered to rats for 14 consecutive days. It was demonstrated to reduce ventricular arrhythmia (VA) score and improve histopathological changes of the myocardium. The underlying cardioprotective mechanism of berberine was caused by decreased p85 activity, a regulatory subunit of PI3K, leading to AKT activation. Suppression of PI3K/AKT signaling pathway is accompanied by a decrease in tumor necrosis factor-⍺ (TNF-⍺), interleukin-6 (IL-6) and IL-1β levels resulting in a reduction in inflammatory cells infiltration. Berberine protected against MIRI through suppression of PI3K/AKT signaling and, subsequently, in the inflammatory response [15].

#### 2.4.5. Dihydroquercetin

Dihydroquercetin (DHQ) is a flavonoid that is commonly found in *Allium cepa* L. and *Silybum marianum* (L.) Gaertn [39]. In H9c2 cells and rat hearts Langendorff perfusion model, DHQ pre-treatment significantly reduced the cardiac dysfunction via scavenging free radical, reduction in lipid peroxidation, and elevation of the antioxidant enzyme activity [31]. Two myocardial enzymes, namely AST and CK, were reduced following DHQ. Western blot revealed that DHQ downregulated CHOP, Caspase-12, and p-JNK. Moreover, DHQ delayed the occurrence of ERS via reduction in GRP78, p-PERK, and p-eif2α expression levels and by up-regulation of HO-1 expression and Nrf2 binding to antioxidant elements. The effects of DHQ were inhibited by PI3K/AKT inhibitor LY294002, which suggest that DHQ alleviated MIRI at a dose of 2.5, 5, 10, 20, 40, and 80 µM via activation of PI3K/AKT pathway, which subsequently reduced oxidative stress and apoptosis induced by ERS [31].

#### 2.4.6. Cortex Dictamni

Cortex Dictamni (CD) is a species of *Dictamnus dasycarpus* Turcz [40,41]. Aqueous extract of Cortex Dictamni (AECD) on MIRI-induced-H92c cells demonstrated improvement of cell viability while reducing LDH and cardiac troponin I (cTn-I) release at a dose of 0.39, 1.56, 6.25, 25, and 100 µg/mL. AECD silenced both the initial and late phase of apoptosis via reduction in nucleus chromatin condensation, phosphatidylserine externalization, and fragmentation of DNA. AECD significantly controlled the oxidative stress via elimination of ROS and MDA while promoting SOD in MIRI. The nrf2-mediated antioxidant pathway was enhanced during AECD pre-treatment. The mitochondrial-dependent apoptosis process was also suppressed by AECD, evidenced by elevation of Bcl-2/Bax ratio, inhibition of cytochrome-c release, and caspase-3 and -9 activation. The anti-apoptotic effects of AECD in MIRI are mainly by activating the PI3K/AKT pathway and GSK-3β phosphorylation. LY294002 inhibited p-GSK-3β and suppressed the Nrf2 expression, which AECD reversed via PI3K/AKT mechanism [13].

#### 2.4.7. *Dracocephalum moldavica*

*Dracocephalum moldavica* L. is a herbaceous plant that belongs to the Lamiaceae family [42]. Total flavonoid extract of *Dracocephalum moldavica* (TFDM) was found to decrease MIRI severity by decreasing CK-MB, LDH, MDA, and apoptotic cells index at aa dose of 3, 10, and 30 mg/kg. Furthermore, it was also able to reduce the infarction size, preserved cell membrane, and mitochondria. In addition, antioxidative and anti-apoptotic properties of TFDM were confirmed by increased SOD and Bcl-2/Bax ratio and decreased cleaved caspase-3, -7 and -9 levels. Additionally, TFDM elevated p-PI3K and p-AKT as well p-PI3K/PI3K and p-AKT/AKT ratio. Activation of PI3K/AKT signaling was facilitated by protein expression of p-GSK-3β and p-ERK1/2. The cardioprotective effect of TFDM was abolished upon administration of LY294002 and PD98059, which were the inhibitor of PI3K and ERK 1/2 phosphorylation, respectively [23].

#### 2.4.8. Tilianin

Tilianin is a powerful flavonoid in dried *Dracocephalum moldavica* L.(DML) [43]. Tilianin at a dose of 2.5, 5, and 10 mg/kg was administered to rats for 14 days. It was found to improve MIRI via promoting cell survival, reducing apoptosis, and scavenging free radical activity. Tilianin reduced CK-MB, LDH, MDA and increased SOD levels. Tilianin inhibited myocardial apoptosis by increasing the Bcl-2/Bax ratio and lowering caspase-7 and -9 levels. X-linked inhibitor of apoptosis protein (XIAP) particularly binds to caspase-3, -7, and -9 and regulates their activities. Following tilianin treatment, Smac/Diablo and HtrA2/Omi protease activity were reduced in the treated-MIRI group, thus suppressing caspases proteolytic activity and improving XIAP degeneration, resulting in the diminished ischemic area with the betterment of myocardial function. Tilianin inhibited apoptosis in MIRI by activating PI3K/AKT signaling and suppressing the mitochondrial leakage of Smac/Diablo and HtrA2/Omi, XIAP degeneration, and caspase activity [22].

#### 2.4.9. Schisandrin B

Schisandrin B (Sch B) is a natural medicinal monomer extracted from *Schisandra chinensis* (Turcz.) Baill. [44] Sch B reduced apoptotic nuclei number, Bax, and cleaved-caspase-3 while increased Bcl-2 at a dose of 60 mg/kg. Furthermore, the significant elevation of p-AKT expression showed that Sch exerted its protective role via PI3K/AKT signaling [24]. AKT activation preserved the mitochondrial integrity via blocking the opening of mPTP and phosphorylation GSK-3β [23,45].

#### 2.4.10. Higenamine

Higenamine derives from plant-based protoberberines alkaloids [46]. Higenamine at a dose of 10 mg/kg increased the cardiomyocytes survival by decreasing the infarct size and apoptotic nuclei. In addition to that, the administration of higenamine at doses of 30, 60, 100, and 120 µM was able to attenuate early and late H_2_O_2_-induced cardiomyocyte apoptosis. The protective effect of higenamine was achieved via upregulation of cleaved-caspase-3 and -9 and protein expression of p-AKT. However, ß_2_-adrenergic receptor (AR) antagonist and PI3K inhibitor eliminated these protective effects [32]. In summary, higenamine protected against MIRI through β2-AR/PI3K/AKT pathway.

#### 2.4.11. *Myrica rubra*

*Myrica rubra* (Lour) Siebold and Zucc. belongs to the family Myricaceae. Hypoxic/reoxygenation of H9c2 cells revealed that *Myrica rubra* flavonoid (MRF), at a dose of 6.25 µg/mL, was able to attenuate intracellular ROS level, protected the cells against mitochondrial membrane potential depolarization, inhibited caspase-3 activity, and alleviated the ratio of apoptotic cells [26]. This protective function of MRF is contributed by the upregulation of Akt and GSK-3ß proteins phosphorylation [26].

#### 2.4.12. Total Paeony Glycoside

Total paeony glycoside (TPG) is a major constituent of *Paeonia lactiflora* Pall. [47]. TPG reduced H9c2 cells apoptotic rate, downregulated the cleaved-caspase-3, PARP1, and Bax, and it upregulated Bcl-2 and pro-caspase-3 at doses of 10, 40, and 160 µg/mL. The anti-apoptotic mechanism of TPG was closely related to its ability to suppress oxidative stress. TPG significantly attenuated the protein expression of PI3K/AKT, which was inhibited by insulin growth factor (IGF-1). These findings suggested that the protective function of TPG was achieved by suppressing the PI3K/AKT mechanism [27].

#### 2.4.13. *Bauhinia championii* Benth

*Bauhinia championii* Benth. or *Phanera championii* Benth. belongs to the genus Bauhinia [48]. *Bauhinia championii* flavone (BCF), at a dose of 3.125 µg/mL, markedly attenuated apoptotic rate, caspase-3, Bax, and cytochrome-C. BCF also significantly increased the protein expression of p-PI3K and p-AKT. In addition, BCF significantly decreased the level of ROS-generating mitochondria, attenuated mitochondrial transmembrane potential (∆Ψm) dissipation, reduced the mPTP opening and increased ATP production. However, co-administration of LY294002 eliminated the protective effect of BCF [28]. Figure 2 shows the chemical structure of the natural products compound that were able to attenuate MIRI through PI3K/AKT signaling pathway.

### 2.5. Techniques to Detect PI3K/Akt Activation and Its Related Mechanism

The studies reported in this review utilized Western blotting analysis for detection of PI3K and Akt. Western blot is not only able to detect the protein expression but also able to quantitate protein phosphorylation, which is the fundamental mechanism to study cell signaling [49]. Analysis of phosphorylation demonstrated the status of the signaling. Most studies in this review measured Akt, PI3K and its phosphorylated proteins. However, some researches only quantified Akt and p-Akt [13,15,24,25,26,29,32]. Akt is the primary protein effector downstream of the PI3K signaling pathway. Some research also measured GSK-3ß, the downstream effector of Akt which participated in mediating apoptosis [13,23,26,29]

Activation of PI3K/Akt signaling involved in the protection against MIRI-induced apoptosis. In order to evaluate the cardioprotective effect played by PI3K/Akt signaling on cell survival, its effects in the apoptotic-related mechanism were observed. There were many assays with various techniques used to measure various aspect of apoptosis. Some studies used flow cytometric quantitation of cell death using Annexin-V/propidium staining to detect early and late apoptotic events. Annexin has greater specificity to bind to the phosphotidylserine (PS), which is the membrane phospholipid. As apoptosis failed to maintain the phospholipid in the cell membrane, PS appear on the outer leaflet of the cell membrane which will be stained with Annexin V. Cells stained positive with Annexin V and negative for PI was suggestive for early apoptotic event, whereas Annexin V positive and PI negative population indicate late apoptotic event [32]. Another study performed terminal deoxynucleotidyl transferase-mediated dUTP-biotin nick end-labeling (TUNEL) assay for detection of apoptotic cells that exhibited the DNA fragmentation. Most of the studies measured the molecular players related to apoptosis/apoptotic cascade such as bcl-2 family protein and caspases activity. Although mitochondria played a key role in apoptosis, only five of the researches measured the parameters related to mitochondrial apoptosis such as mitochondrial membrane potential, mPTP opening, and release of cytochrome-c from mitochondria to cytosol [13,14,25,28,31]. The cardioprotective effect of NP against MIRI on cell survival was measured by regulation of PI3K/AKT signaling and its effects in the apoptotic-related mechanism (Table 4).

### 2.6. PI3K Inhibitor

The antagonizing effect of PI3K/Akt inhibitors against NP were important to be verified to confirm the PI3K/Akt signaling pathway involvement. Protective effects of NPs in this review were abolished upon administration of PI3K inhibitors such as LY 294002, LY 204002, and wortmannin. There were two studies that did not use PI3K inhibitor [15,27] (Table 4). PI3K inhibitor caused the rapid induction in cell death. Therefore, the incubation of cell line with PI3K/Akt inhibitors inhibited the effect of NPs on cellular apoptotic rate, reversed the expression of pro- and anti-apoptotic molecules, and eliminated the anti-apoptotic effect of NPs.

## 3. Discussion

This scoping review suggests the evidence of the NP-based therapies that provide cardioprotection in MIRI targeting the PI3K/AKT mechanism. This became a novel therapeutic strategy in the treatment of MIRI. There are increasing numbers of NP-derived drugs used in medicine due to their various biological activities and drug-like properties. One of the major metabolites in the plant is a flavonoid. Previous studies showed that a high intake of dietary flavonoids and their subclasses were associated with lower mortality of cardiovascular disease [50]. The NPs also enriched with other secondary metabolite compounds such as alkaloid and saponin, which display cardioprotective function [14,30].

MIRI is associated with multiple pathological condition such as oxidative stress, ERS, mitochondrial injury, inflammatory process, and apoptosis. Therefore, the role of PI3K/Akt signaling pathway post MIRI is very crucial, as it improves the cell survival and viability by limiting the apoptotic event. Generally, the apoptotic signaling pathway can either be stimulated by intrinsic or extrinsic pathways [51]. Bcl-2 and Bax proteins are integral regulatory factors in all apoptotic pathways [51,52]. The intrinsic pathway mainly involves mitochondrial injury and caspase-9 activation, whereas the extrinsic pathway is mediated by the death receptor Bcl-2/Bax [51,52]. Another pathway is regulated through the activation of caspase-12 [53]. Caspase-9 activates caspases-3 and -7, causing the conformation and biochemical processes related to apoptosis [51]. Apoptotic cascade includes mitochondria dysfunction, disturbed ATP supply, loss of mitochondrial membrane potential, the opening of mPTP, and release of cytochrome-C [54,55]. Therefore, inhibition of apoptosis and oxidative stress by targeting Bcl-2/Bax, cytochrome-C, caspase-3, -7, -9, mitochondrial ROS attenuated the progression of MIRI.

ERS is a type of oxidative stress which induces cell dysfunction in MIRI via regulation of the PI3K/AKT pathway [56]. Oxygen deficit and ROS accumulation during MIRI can trigger unfolded protein response (UPR) [56,57], through three ER transmembrane sensors: PERK, Activating Transcription Factor 6 (ATF6), and Inositol requiring 1 (IRE1). Uncontrolled ERS dissociates GRP78 from ER transmembrane sensors [58]. Inhibition of caspase-12 prevents subsequent caspase-9 and -3 activation and eventually suppressing apoptosis, and promoting cell survival [14,58]. Failure to control ERS might lead to apoptosis via increased GRP78, CHOP pathway, and caspase-12 activation, therefore dysregulating the PI3K/AKT signaling leading to MIRI.

Nrf2 is a redox-sensitive transcription factor that plays vital role to combat oxidative stress [59]. Nrf2 coordinates transcriptional activation of antioxidant genes including HO-1, NQO1 and SOD to eliminate ROS in reperfused cardiomyocytes [60,61]. Modulation of PI3K/AKT signaling pathway is important to increase SOD, CAT, and GSH enzymes activity and reduce MDA value in MIRI cardiomyocytes [62,63]. Previous studies demonstrated that attenuating the inflammation is critical for protection against MIRI [64] via PI3K/AKT pathway. The inhibition of PI3K/AKT pathway reduced IL-6, IL-1β, and TNF⍺ in serum and histopathological analysis [15]. The PI3K/AKT pathway is believed to regulate the adaptive immune response and innate immune cells and suppress the inflammation associated with MIRI [65].

PI3K/AKT pathway is a key signaling cascade that mitigates MIRI. Most of NPs in Figure 1 conferred cardioprotection against MIRI via activation of PI3K/AKT pathway [13,14,22,23,24,26,28,29,30,31,32] except berberine [15] and TPG (41), which suppressed PI3K/AKT pathway. Further phosphorylation of downstream effectors of AKT, GSK-3β, involved cell survival in mitigating MIRI [13,23,26,29,66] by inhibiting mPTP opening, protecting the cell against mitochondrial-mediated cell death, and limiting infarct size [23]. β2-AR agonist activated PI3K/AKT cascade to promote myocyte survival [32]. The phosphorylation of PI3K/AKT pathway to mitigate MIRI was markedly reversed by a specific PI3K inhibitor, eliminating the unwanted effects on cytochrome-C release, caspase-9/3 activity, Bcl-2/Bax ratio, mPTP opening, ERS, p53 level, and XIAP expression. These findings conclude the key role of the PI3K/AKT pathway to suppress MIRI by attenuating ERS, oxidative stress, inflammation, and apoptosis [29,30,31]. The mechanism underlying NPs alleviates MIRI through activation of PI3K/Akt pathway is summarized in Figure 3.

## 4. Materials and Methods

A systematic search was conducted to identify the NP that alleviate MIRI targeting on PI3K/AKT pathway using PRISMA framework for scoping review [67]. Peer-reviewed and English full-text articles were gathered from January 2016 to January 2021 using PubMed, Scopus, and Web of Science (WOS) databases. The search terms used together with the Boolean operators AND and OR were as follows: “natural product” OR “herb* medicine” OR “plant-derived” OR “natural medicine” OR “plant extract” OR “plant ground” AND “myocardial injury” OR “myocardial infarction” OR “myocardial damage” OR “myocardial isch*” OR “myocardial inflammation” OR “myocardial dysfunction” AND “PI3K/AKT”. The inclusion criteria of this study were (1) targeting on PI3K/AKT signaling, (2) availability of English full text, and (3) experiment using MIRI model (Table 5). The exclusion criteria were (1) a combination of herbs in the treatment and (2) not plant-derived. All titles and abstracts from the search results were screened and critiqued independently by all authors. Following the inclusion and exclusion criteria, the articles were shortlisted for eligibility upon reaching a consensus by all authors.

## 5. Limitations

The findings from this scoping review were restricted by several factors. First, our search strategy was focused solely on targeting one signaling pathway, namely the PI3K/Akt pathway, while MIRI is a complex injury that involves various other pathways such as TGFβ1/Smads signaling pathway [68], Parkin-mediated mitophagy [69], protein kinase A pathway [70], and activation of endogenous pro survival pathways such as protein kinase G [71]. Therefore, targeting one mechanism at a time may not be sufficient to produce a strong and robust effect in clinical situations where many uncontrolled variables usually coexist. Second, the therapeutic use of natural products was only limited to pre-clinical studies with no clinical trials included. Although all animal models in the papers presented were rodents, there was a difference in the genus. Additionally, we used a small sample size, which may have limited the generalizability of our findings. Third, several of the included studies provided experimental values in graphic form and, thus, the authors were unable to compare the findings in one study to another study that measured the same parameter. Fourth, translation into clinical situations is unfeasible because myocardial infarction is unpredictable, and most of the NPs-based drugs were designed as a pretreatment in MIRI models. Finally, only English articles with accessibility of full text were included. However, given the limitations faced in conducting this scoping review, it is unlikely that any missed data would possibly amend the conclusion drawn based on this review due to a clear focus on pre-clinical studies and because the electronic database search was performed to include citations and anywhere the keywords appeared in the article.

## 6. Conclusions

We summarized that NPs enriched with various bioactive compounds are beneficial to mitigate the pathological effects in post-MIRI. Recent studies suggest that these harmful effects can be alleviated by targeting the PI3K/AKT pathway, which might reduce apoptosis, inflammation, and oxidative stress. Further research is warranted to validate the safety and efficacy of NPs-based drugs in MIRI.

## Figures and Tables

**Figure 1 pharmaceuticals-16-00739-f001:**
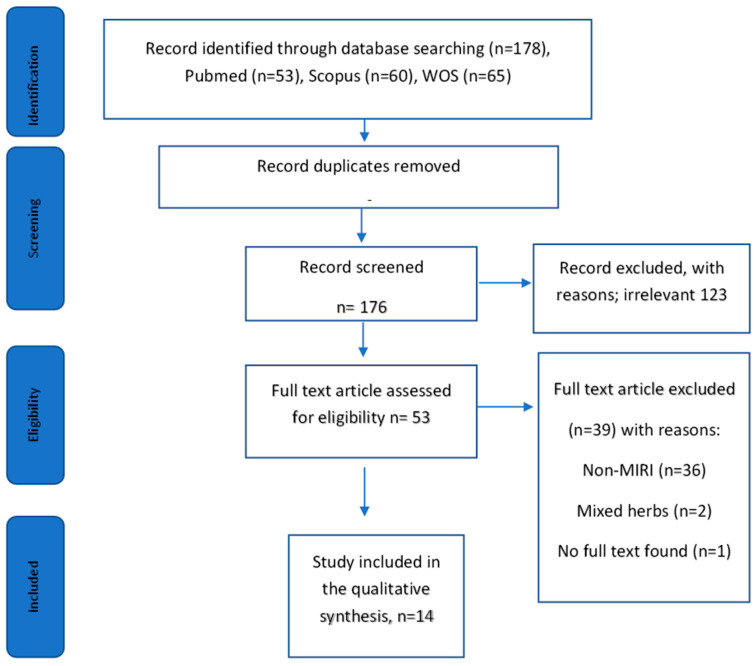
Flow diagram of the article selection.

**Figure 2 pharmaceuticals-16-00739-f002:**
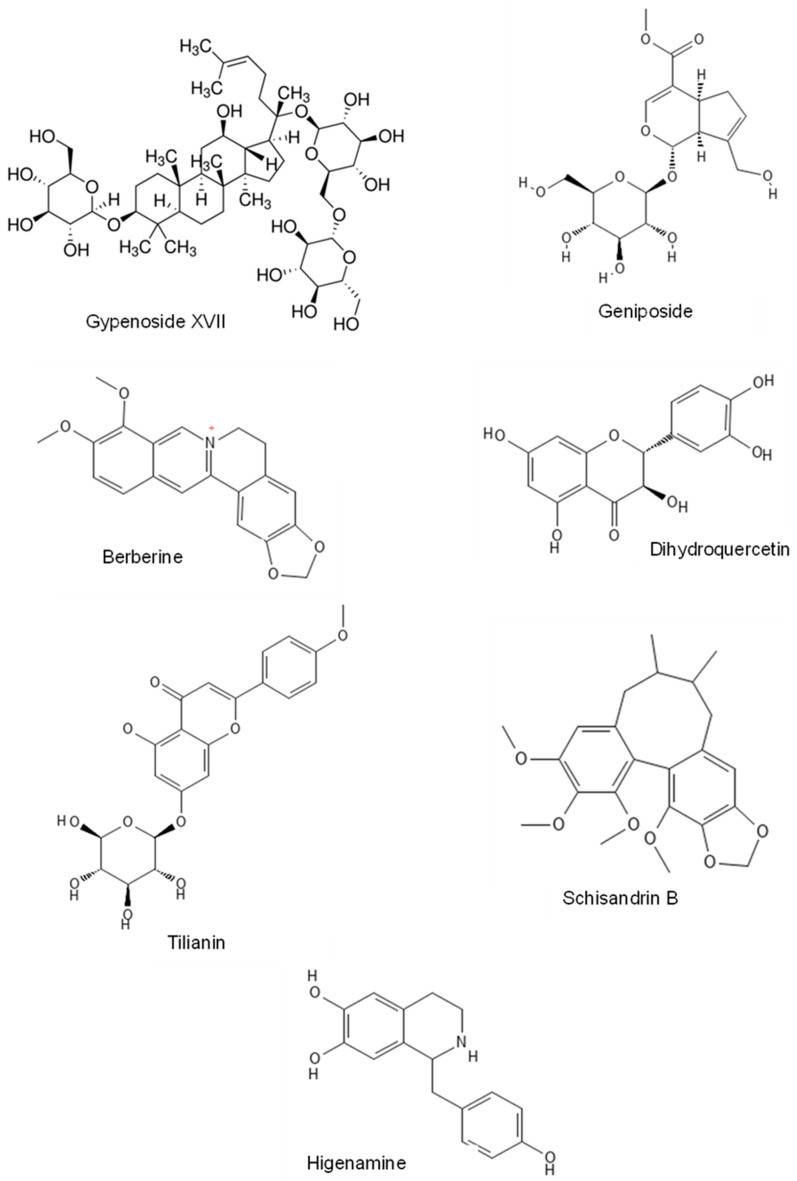
Chemical structures of natural products compound which attenuate MIRI through PI3K/AKT signaling.

**Figure 3 pharmaceuticals-16-00739-f003:**
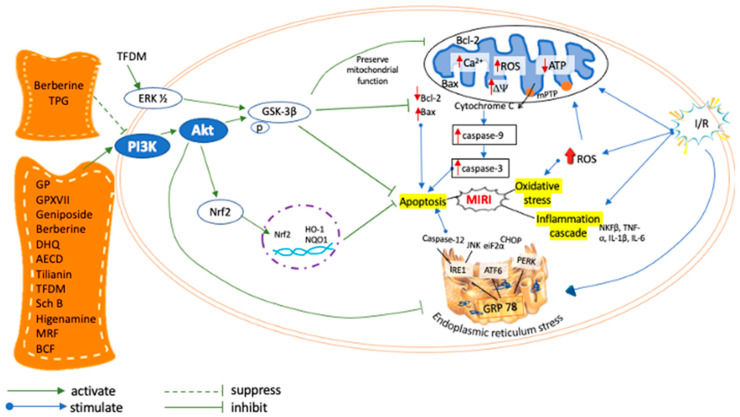
Mechanism of action of NPs alleviates MIRI in cardiomyocytes through PI3K/Akt signaling pathway.

**Table 1 pharmaceuticals-16-00739-t001:** Study characteristics of the selected articles.

Study Characteristics		Count *n* (%)
Year of publication	2016	5 (35.7%)
	2017	3 (21.4%)
	2018	3 (21.4%)
	2019	3 (21.4%)
Country of publication	China	14 (100%)
Source of plant	Part of the plant, method of extraction and the solvents used were mentioned.	5 (35.7%)
	Processed plants were purchased.	5 (35.7%)
	Not mentioned source of plant	4 (28.6%)
MIRI model used	In vitro	5 (35.7%)
	In vivo	4 (28.6%)
	Mixed in vitro and ex vivo	2 (14.3%)
	Mixed in vitro and in vivo	2 (14.3%)
	In vivo, in vitro, ex vivo	1 (7.1%)

**Table 2 pharmaceuticals-16-00739-t002:** MIRI models used and method to inflict MIRI.

MIRI Model Used	Brief Explanation of Method to Inflict MIRI	Duration of Ischemic/Reperfusion (I/R)
In vitro	Hypoxia-reperfusion (H/R) injury was elicited in H9c2 cardiomyocytes. To induce hypoxia, the cells were incubated in an anaerobic box with or without the media changed to non-glucose DMEM. To mimic reperfusion, the cells were moved to the regular incubator with or without the media being replaced with high-glucose media. Duration of ischemia and reperfusion varies according to the studies.	4 h of hypoxia, 24 h of reperfusion [29]
		6 h of hypoxia and 12 h of reoxygenation [14]
		H/R time: 4/2, 6/3, 12/4, 14/5, 16/6, 22/10 h [25]
		6 h of ischemia, 16 h of reperfusion [31]
		8 h of hypoxia followed by 4 h of reoxygenation [13]
		6 h of ischemia, followed by reoxygenation [26]
		10 h of ischemia, followed by 2 h of reperfusion [27]
		6 h of hypoxia, followed by 12 h of reoxygenation [28]
In vivo	Left anterior descending coronary arteries (LAD) of MIRI rat model were ligated temporarily to induce ischemia, followed by a period of reperfusion by releasing the ligation. Duration of ischemia and reperfusion varies according to the studies.	LAD were reversibly occluded for 45 min, followed by 3 h of reperfusion [29]
		LAD was occluded for 30 min followed by reperfusion for 4 h [15]
		45 min of ischemia followed by 4 h of reperfusion [22]
		Ischemia for 45 min, followed by 12 h of reperfusion [23]
		45 min ischemia, followed by 24 h of reperfusion [24]
		Ligation LAD for 30 min followed by 24 h of reperfusion [32].
	Cold ischemia was performed to induce ischemia followed by reperfusion.	Cold ischemia for 18 h, followed by 8 h of reperfusion [30]
Ex vivo	Langendorff perfusion of rat hearts was used to induce global ischemia and reperfusion.	Global ischemia for 40 min and reperfusion for 60 min [14].
		45 min of ischemia, 1 h reperfusion [31]
		Ischemic state was induced for 30 min by no-flow followed by 30 min of reperfusion [32].

**Table 3 pharmaceuticals-16-00739-t003:** Source of plants, dose, and duration of intervention.

Natural Product	Source	Dose and Duration	References
Gypenoside (GP)	*Gynostema pentaphyllum* (Thunb.) Makino	For in vivo Short term: GP 50, 100, 200 mg/kg 1 h before MIRI. Long term: GP 200 mg/kg 1 h before MIRI; 24 and 72 h post-MIRI.	[29]
		For in vitro GP 5µM, 10µM, 20µM incubated before OGD/R	
Gypenoside XVII (GP-17)	Antioxidant phytoestrogen of the gypenoside group.	For in vitro GP-17 1.25, 2.5, 5, 10, 20, 40, 100 µM incubated 24 h before H/R injury.	[14]
		For ex vivo GP-17 5, 10, 20 µM were dissolved in Krebs–Henseleit (KH) buffer	
Geniposide (glycoside)	An extract from *Gardenia jasminoides* J. Ellis	For in vitro Geniposide 2.5, 5, 10, 20, 40, 80, 160, 320 µM for 30 min before H/R	[25]
Berberine	An alkaloid derivative. The source of plant was not mentioned	For in vivo 10 mg/kg berberine OD for 30 days	[30]
		For in vitro 10 mg/mL berberine for 24 h	
Berberine	The source of plant was not mentioned	For in vivo Berberine 25, 50, 100 mg/kg for 14 days	[15]
Dihydroquercetin (DHQ)	Belongs to the flavanonol subclass in the flavonoids. The source of plant was not mentioned	For in vitro DHQ 2.5, 5, 10, 20, 40, 80 µM incubated for 12 h before H/R	[31]
Aqueous extract of Cortex Dictamni (AECD)	A species of Dictamnus dasycarpus Turcz. This plant was aqueously extracted.	For in vitro AECD 0.39, 1.56, 6.25, 25, 100 µg/mL were pretreated with the cells for 24 h.	[13]
Tilianin (is a flavonoid antioxidant)	Tilianin is a powerful flavonoid from dried *Dracocephalum moldavica* L. (DML). Its dried powder was extracted in aqueous ethanol.	For in vivo Tilianin 2.5, 5, 10 mg/kg for 14 days	[22]
Total flavonoid extract from *Dracocephalum moldavica* L. (TFDM)	The plant was extracted in 40% ethanol andHPLC was preformed	For in vivo TFDM 3, 10, 30 mg/kg for 2 weeks	[23]
Schisandrin B. (Sch B)	Natural monomer extracted from *Schisandra chinensis* (Turcz.) Baill.	For in vivo Sch. B 60 mg/kg for 15 days	[24]
Higenamine	An alkaloid derivative. The source of plant was not mentioned	For in vivo Higenamine 10 mg/kg 2 h before surgery	[32]
		For in vitro Higenamine 30, 60, 120 µM	
		For ex vivo Higenamine 100µM	
Myrica rubra flavonoids (MRF)	The bark of Myrica rubra was extracted with methanol using reflux extraction. Myrica rubra flavonoid was then chemically identified at the Institute of Medicinal Plant Development.	For in vitro MRF 6.25µg/mL was incubated for 12 h before H/R injury	[26]
Total paeony glycoside (TPG)	TPG powder was obtained from Haoxuan Biological Technology Co., Ltd. (Xi’an, China).	For in vitro TPG 10, 40, 160 µg/mL post I/R injury	[27]
Bauhinia championii flavone	Dried Bauhinia championii (Benth.) Benth. was extracted with ethanol.	For in vitro BCF 3.125 µg/mL	[28]

**Table 4 pharmaceuticals-16-00739-t004:** Summary of regulation of PI3K/Akt by NPs to alleviate MIRI and its possible mechanism.

Natural Products	Regulation of PI3K/Akt Signaling	Administration of PI3K Inhibitor	Parameters Used to Measure Cardioprotective Mechanism of NPs on Cell Survival					References
			Apoptosis	Mitochondrial Injury/Cytochrome C/Caspase-3,9	Bcl-2/Bax	Endoplasmic Reticulum Stress Marker	Oxidative Stress/Antioxidant	
Gypenoside (GP)	↑p-Akt/Akt, p-GSK3β/GSK3β	√	√ (TUNEL)	×	√	√	×	[29]
Gypenoside XVII (GP-17)	↑p-PI3K/PI3K, p-Akt/Akt ratios	√	√ (Annexin V/PI)	√	√	√	√	[14]
Geniposide (glycoside)	↑p-Akt/Akt	√	√ (Annexin-V/PI)	√	√	×	√	[25]
Berberine	↑expression of PI3K, Akt, p-Akt	√	√	√	√	×	×	[30]
Berberine	↓p-Akt/Akt ratio ↓p-p85/p85	×	×	×	×	×	×	[15]
Dihydroquercetin (DHQ)	↑p-PI3K/PI3K and p-AKT/AKT ratios	√	×	√	√	√	√	[31]
Aqueous extract of Cortex Dictamni (AECD)	↑p-Akt, p-GSK-3β	√	√ (TUNEL, Annexin/PI)	√	√	×	√	[13]
Tilianin (is a flavonoid antioxidant)	↑p-Akt, p-PI3K, p-Akt/Akt, p-PI3K/PI3K	√	√ (TUNEL)	√	√	×	√	[22]
Total flavonoid extract from *Dracocephalum moldavica* L. (TFDM)	↑protein expression of p-PI3K, p-Akt ↑p-PI3K/PI3K, p-Akt/Akt ratios ↑p-GSK-3β, p-ERK 1/2	√	√ (TUNEL)	√	√	×	√	[23]
Schisandrin B. (Sch B)	↑p-Akt	√	√ (TUNEL)	√	√	×	×	[24]
Higenamine	↑p-Akt, p-Akt/Akt ratio	√	√	√	×	×	×	[32]
Myrica rubra flavonoids (MRF)	↑p-Akt/Akt, p-GSK-3β/GSK-3β	√	√ (Annexin V/PI)	√	√	×	√	[26]
Total paeony glycoside (TPG)	↓p-PI3K, p-Akt	×	√ (Annexin V-FITC/PI)	√	√	×	√	[27]
Bauhinia championii flavone	↑p-Akt/Akt, p-PI3K	√	√ (Annexin V-FITC/PI)	√	√	×	√	[28]

**Table 5 pharmaceuticals-16-00739-t005:** Eligibility criteria for article selection.

Eligibility Criteria	Criteria
For title selection	1. Title in English 2. Year of publication from January 2016 to January 2021 3. Title reflects on MIRI
For abstract selection	1. Abstract reflects that the article is an original article. 2. Abstract provides evidence of a robust study design. 3. Abstract highlights at least one measurable outcome that reflects PI3K/Akt signaling pathway. 4. Abstract clearly mentions the usage of MIRI model. The abstract reflects the use of natural product as intervention.
For full text of article selection	1. The article is available as a full-text article. 2.The article provides well-designed research methodology and/or intervention. 3. The article measures the outcome that related to PI3K/Akt. 4. The article uses natural product as an intervention. 5. The natural product used as intervention was not a combination of multiple natural products. The article explicitly explains the usage of MIRI model.

## Data Availability

Data is contained within the article.

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
