# Peer review of "Natural Products Targeting PI3K/AKT in Myocardial Ischemic Reperfusion Injury: A Scoping Review"

_pharmaceuticals, 2023, doi:10.3390/ph16050739_

Round 1

Reviewer 1 Report

The authors have done a thorough review of the work and is written very systematically and brings out the importance of natural product derivatives in cardio-protection against ischemia-reperfusion injury. The selected papers for the review of natural compounds are relevant and their effect on PI3K/Akt pathway is clearly and descriptively discussed. It would be interesting to check whether any studies show binding (binding kinetics) studies of these compounds with the components of PI3K/Akt pathway and include a discussion on that as well. This would clearly establish a direct effect of these compounds on the PI3K/Akt pathway. Additionally, do these compounds in the review have any common structural motif that makes them to be classified to be activators of PI3K/Akt pathway? It seems like they have identical chemical structure with few variations on them. Can there be a chemical structural analysis included as a discussion?

Author Response

Pharmaceuticals - 2300212

Type of manuscript: Review

Title: Natural products targeting PI3K/AKT in myocardial ischemic reperfusion

injury: A Scoping Review

Response to reviewers

Dear Editor,

Thank you for giving us the opportunity to submit a revised draft of the manuscript “Natural products targeting PI3K/AKT in myocardial ischemic reperfusion injury: A Scoping Review” for publication in the Pharmaceuticals Journal. We appreciate the time and effort that you and the reviewers dedicated for providing feedback on our manuscript and are grateful for the insightful comments on and valuable improvement to our paper. We have incorporated most of the suggestions made by the reviewer, and those changes are highlighted in yellow within the manuscript. Please see below, in blue, for a point-by-point response to the reviewers’ comments and concerns. All line numbers refer to the revised manuscript file.

Reviewer 1

The authors have done a thorough review of the work and is written very systematically and brings out the importance of natural product derivatives in cardio-protection against ischemia-reperfusion injury. The selected papers for the review of natural compounds are relevant and their effect on PI3K/Akt pathway is clearly and descriptively discussed.

It would be interesting to check whether any studies show binding (binding kinetics) studies of these compounds with the components of PI3K/Akt pathway and include a discussion on that as well. This would clearly establish a direct effect of these compounds on the PI3K/Akt pathway.

We thank reviewer for the comments.

We have run a thorough literature search and there few literatures which showed binding kinetics with P13K/AKT.  However, as these studies were not in the scope of MIRI, we excluded the literature in accordance to our exclusion criteria. Kindly refer to table below for further reason for exclusion. Therefore, we would not include the binding kinetics as part of the discussion. We appreciate reviewer understanding in this matter.

Natural product

Model

Link

Remark

Gypenoside

Bladder Cancer

https://www.ncbi.nlm.nih.gov/pmc/articles/PMC8984741/

Exclude d/t not in the scope of MIRI

renal cell carcinoma

https://pubmed.ncbi.nlm.nih.gov/33556477/

Geniposide

Neuropathic Pain

https://pubmed.ncbi.nlm.nih.gov/35699893/

epilepsy

https://www.ncbi.nlm.nih.gov/pmc/articles/PMC5772842/

Berberine

colon cancer cells

https://www.ncbi.nlm.nih.gov/pmc/articles/PMC7808376/

Dihydroquercetin

-

-

No binding kinetic study

Aqueous extract of Cortex Dictamni (AECD)

-

-

Tilianin (is a flavonoid antioxidant)

-

-

Total flavonoid extract from Dracocephalum

-

-

Moldavica L. (TFDM)

-

-

Schisandrin B. (Sch B)

-

-

Higenamine

-

-

Myrica rubra flavonoids (MRF)

-

-

Total paeony glycoside (TPG)

Fatty Liver Disease

https://www.ncbi.nlm.nih.gov/pmc/articles/PMC8481579/

Exclude d/t not in the scope of MIRI

Bauhinia championii flavone

-

-

No binding kinetic study

Additionally, do these compounds in the review have any common structural motif that makes them to be classified to be activators of PI3K/Akt pathway? It seems like they have identical chemical structure with few variations on them. Can there be a chemical structural analysis included as a discussion?

We thank reviewer for the comments. We consulted a chemist specialized in natural products and by referring to the chemical structures in Figure 2, we are not able to compare the common structural motif of our natural compounds. This is because these compounds are not in the same class and therefore, we could not do structure relationship activity study as requested. We hope that reviewer could accept our justification in this matter.

Reviewer 2 Report

Manuscript ID: pharmaceuticals-2300212
Type of manuscript: Review
Title: Natural products targeting PI3K/AKT in myocardial ischemic reperfusion
injury: A Scoping Review
Authors: Syarifah Aisyah Syed Abd Halim, Norhashima Abd Rashid, Choy Ker Woon, Nahdia Afiifah Abdul Jalil - Submitted to section: Natural Products

Dear Authors,

I have read and reviewed the above-mentioned scoping review with interest and pleasure.

Here are my comments and recommendation:

0. Title & Abstract

Line 2: The authors should perhaps delite the term "myocardial" in their title, because a) IRI is not restricted to the myocardial vasculature; and b) this review may reach a broader readership when ischemia-reperfusion injury is seen in a bigger picture. Perhaps change "MIRI" into "IRI". See my comments below.

1. Introduction 

Line 37: Authors pretty well explained MIRI and however wrote in the Introduction: "Ironically, the perfusion process itself can be harmful... MIRI [3]." To underline the broad importance of this review, authors should also mention that ischemia-reperfusion injury is not restricted to the myocardial vasculature, i.e. ischemia-reperfusion injury may be even more harmful in the cerebral tissue particularly in cases blood-brain-barrier breakdown. Therefore, authors should change this passage into the following: "Ironically, the perfusion process itself can be harmful... MIRI [3]. Moreover, similar ischemia-reperfusion phenomena are known in many vascular territories of different organs, e.g., in the brain leading to blood-brain-barrier breakdown and secondary brain swelling  [cite here Liebeskind et al Stroke 2019]." Furthermore, please change the next sentence into the following: "The damage includes fulminant migration of inflammatory cells, oedema and apoptotic processes, an extension of infarct area and dysfunctional microcirculation [4; cite here Michel and Curry Physiol Rev 1999; Bosche et al Stroke 2003, Dohmen et al Stroke 2007]."

2. Results

What about the PRISM criteria for review processes. Did the authors follow this guidlines? If the authors did not use this criteria, authors should add (at the end of the Introduction (Line 62-64), or in the Result section (line 73), or in the Material and Method/Discussion as a limitation the following: "As our article serves as a scoping review, PRISM criteria were not be used."

Figure 1. Authors should explain what did they mean by: "Record excluded, with reasons; irrelevant 123". This needs further clarification, in my view.

In General, the table containing important information should be depicted in a more condence form. It is really hard to get all information. Please present the table in a different form to provide a better overview. Moreover, there is perhaps a line to much according to [ref. 25] in the right column. Or, is there a missing ref according to 2. In vitro? Please may correct this issue.

3. Discussion 

Line 369, Authors should provide a wider picture of apoptotic signalling pathoways including further refs. Please change as follows: "Generally, the apoptotic signally pathways ... [49; please cite here Benson & Trabak Cell Calcium 2023; Bentz et al Cell Physiol & Biochem 2010]."  

Line 410, "Further phosphorylation ... GSK-3ß, involved cell survival in mitigating (M)IRI [10, 11, 18-20, 22,24-28, cite here Bosche et al Prog. in Neuro-Psychopharm. & Biol. Psychiatry 2016; Haupt et al Neuropharmacology 2020; Haupt et al. Stem cells translational medicine 2021]."

4. Material and Methods

Please compare first issue of the Result 

5. Conclusion

Line 451, Please change the first sentence of the Conclusion as follows: "We summarized that ... are benifical to to mitigate the pathophysiological effects in post-MIRI and related pathophysiologic sequilae in other tissue types." 

Line 456, Please change as follows: Further research is warrented to validate the safety and efficacy of NPs-based drugs in ischemia-reperfusion injury."

Kind regards 

Your reviewer

Author Response

Pharmaceuticals - 2300212

Type of manuscript: Review

Title: Natural products targeting PI3K/AKT in myocardial ischemic reperfusion

injury: A Scoping Review

Response to reviewers

Dear Editor,

Thank you for giving us the opportunity to submit a revised draft of the manuscript “Natural products targeting PI3K/AKT in myocardial ischemic reperfusion injury: A Scoping Review” for publication in the Pharmaceuticals Journal. We appreciate the time and effort that you and the reviewers dedicated for providing feedback on our manuscript and are grateful for the insightful comments on and valuable improvement to our paper. We have incorporated most of the suggestions made by the reviewer, and those changes are highlighted in yellow within the manuscript. Please see below, in blue, for a point-by-point response to the reviewers’ comments and concerns. All line numbers refer to the revised manuscript file.

Reviewer 2

Here are my comments and recommendation:

  1. Title & Abstract

Line 2: The authors should perhaps delite the term "myocardial" in their title, because a) IRI is not restricted to the myocardial vasculature; and b) this review may reach a broader readership when ischemia-reperfusion injury is seen in a bigger picture. Perhaps change "MIRI" into "IRI". See my comments below.

We thank reviewer for the comments. We agree that ischemic perfusion injury is not only confined in the myocardium and can occur at other organ such as brain and kidney. However, in this scoping review, our focus was to look for the effects of the natural products on the cardiovascular system and particularly MIRI. And, that is why the search strategy was designed to focus on our main objective.

  1. Introduction 

Line 37: Authors pretty well explained MIRI and however wrote in the Introduction: "Ironically, the perfusion process itself can be harmful... MIRI [3]." To underline the broad importance of this review, authors should also mention that ischemia-reperfusion injury is not restricted to the myocardial vasculature, i.e. ischemia-reperfusion injury may be even more harmful in the cerebral tissue particularly in cases blood-brain-barrier breakdown. Therefore, authors should change this passage into the following: "Ironically, the perfusion process itself can be harmful... MIRI [3]. Moreover, similar ischemia-reperfusion phenomena are known in many vascular territories of different organs, e.g., in the brain leading to blood-brain-barrier breakdown and secondary brain swelling  [cite here Liebeskind et al Stroke 2019]." Furthermore, please change the next sentence into the following: "The damage includes fulminant migration of inflammatory cells, oedema and apoptotic processes, an extension of infarct area and dysfunctional microcirculation [4; cite here Michel and Curry Physiol Rev 1999; Bosche et al Stroke 2003, Dohmen et al Stroke 2007]."

Thank you for the comment. We really appreciate it. We focus on collecting evidence on the natural product's efficacy in alleviating MIRI. Therefore, the relevant information regarding MIRI and its possible mechanisms was discussed in the introduction. We also accept the reviewer’s suggestion to change the sentence in lines 39 & 40; the suggested citation was included between lines 37-49.

“Ironically, the perfusion process can be harmful by intensifying the

tissue injury known as myocardial ischemia-reperfusion injury (MIRI) [3] . Ischemia-

Reperfusion injury refers to the situation where the restoration of the blood flow to the

particular tissues that were previously deprived of oxygen paradoxically worsens their

function and leads to cellular death. Although the re-establishment of blood flow is

necessary to save the ischemic tissue, the reperfusion process could lead to additional

damage, endangering the survival and function of the organ. Ischemic-reperfusion injury

is not only limited to the myocardium but also affects other organs such as the brain,

kidney, gut, and skeletal muscle depending on the impaired blood vessel [4–6] . The

damage includes fulminant migration of inflammatory cells, oedema and apoptotic

processes, an extension of the infarct area and a dysfunctional microcirculation The

damage includes fulminant migration of inflammatory cells, oedema and apoptotic

processes, an extension of the infarct area and a dysfunctional microcirculation [4,7] .”

  1. Results

What about the PRISM criteria for review processes. Did the authors follow this guidelines? If the authors did not use this criteria, authors should add (at the end of the Introduction (Line 62-64), or in the Result section (line 73), or in the Material and Method/Discussion as a limitation the following: "As our article serves as a scoping review, PRISM criteria were not be used."

We thank the reviewer for the comment. We already mentioned that we follow the PRISMA framework in our method (line 448-449 & figure 1).  The reference is included, and the table of eligibility criteria (Table 5) is also added to the paper (line 462).

Figure 1. Authors should explain what did they mean by: "Record excluded, with reasons; irrelevant 123". This needs further clarification, in my view.

We thank the reviewer for the comment. The amendment can be found between lines 80-85.

“The reasons for the irrelevance are non-MIRI models (diabetic cardiomyopathy, myocardial fibrosis, cardiotoxicity and myocardial infarction, high glucose-induced injury in cardiomyocytes, inorganic mercury-induced cardiac injury), measuring other signaling pathways such as PI3-kinase/glycogen synthase kinase 3β/β-catenin pathway, NFkB signaling, IGF-1R/Nrf2 signaling and the PI3k/Akt signaling pathway were not measured.”

In General, the table containing important information should be depicted in a more condence form. It is really hard to get all information. Please present the table in a different form to provide a better overview. Moreover, there is perhaps a line to much according to [ref. 25] in the right column. Or, is there a missing ref according to 2. In vitro? Please may correct this issue.

Thank you for the comment.  Previously, we received comments from reviewers to make the table more comprehensive. Hence, the table presented contains a lot of information as we amended it according to the reviewers’ recommendations. To address this comment, we have come up with 4 different tables to make it more understandable to the readers (Tables 1, 2, 3 and 4).

Table 1: Study characteristics of selected articles (line 94)

Table 2: MIRI model used in the studies and method to inflict MIRI (line 134)

Table 3: Source of plant, dose and duration of intervention (line 145)        

Table 4:  Summary of the effect of natural products on the regulation of PI3K/Akt signaling to alleviate MIRI and its possible mechanism. (line 364)

  1. Discussion 

Line 369, Authors should provide a wider picture of apoptotic signalling pathoways including further refs. Please change as follows: "Generally, the apoptotic signally pathways ... [49; please cite here Benson & Trabak Cell Calcium 2023; Bentz et al Cell Physiol & Biochem 2010]."  

Line 410, "Further phosphorylation ... GSK-3ß, involved cell survival in mitigating (M)IRI [10, 11, 18-20, 22,24-28, cite here Bosche et al Prog. in Neuro-Psychopharm. & Biol. Psychiatry 2016; Haupt et al Neuropharmacology 2020; Haupt et al. Stem cells translational medicine 2021]."

We thank the reviewer for the suggestion. We have included the citation where it is relevant between lines 663-665. The articles that are very specific to traumatic brain injury, we are not able to cite, as our main focus is ischemic reperfusion injury in the myocardium. We hope the reviewer can accept our justification.

  1. Material and Methods

Please compare first issue of the Result 

Thank you for the comment. We already explained above that we strictly follow PRISMA guidelines for Scoping Review. We improved the material & method section (between lines 447 and 461) and included the relevant citation and table.

  1. Conclusion

Line 451, Please change the first sentence of the Conclusion as follows: "We summarized that ... are benifical to to mitigate the pathophysiological effects in post-MIRI and related pathophysiologic sequilae in other tissue types." 

Line 456, Please change as follows: Further research is warrented to validate the safety and efficacy of NPs-based drugs in ischemia-reperfusion injury."

We thank reviewer for the comments. However, we would prefer the manuscript to be more specific to MIRI as we would like to focus on the effect of these natural compounds on the cardiovascular system.

Reviewer 3 Report

The manuscript entitled: “Natural products targeting PI3K/AKT in myocardial ischemic in reperfusion injury: A Scoping Review” by Syarifah Aisyah Syed Abd Halim et al.  is a scoping review aimed to summarize the studies that evaluated the impact of natural products of the PI3K/AKT pathway in ischemia-reperfusion injury. The Authors evaluated 14 papers, that analyzed 10 natural compounds, by using different methodological approach. In general, this is well written review on an interesting topic. However, in my opinion it should be improved.

Major

A)   The Authors should consider to change the title of their manuscript since the majority of the studies that they analyzed were in vitro observations. Therefore, the term “myocardial” is not appropriate, because it is referred to the whole heart that include different cell types.

B)   The Authors should stress the limitations of their analysis, underlying the paucity of the available studies in this area, the different experimental models used into the studies, and the heterogeneity of the compounds tested.  

Minor

A)   Since the main topic of this paper is the ischemia/reperfusion injury, and not the myocardial infarction, the first seven lines of the introduction (from line 31 to 37) can be erased.

B)   A brief comment on the beneficial effect of the activation of the PI3K/AKT pathway in ischemia-reperfusion injury should be included in the introduction.

Author Response

Pharmaceuticals - 2300212

Type of manuscript: Review

Title: Natural products targeting PI3K/AKT in myocardial ischemic reperfusion

injury: A Scoping Review

Response to reviewers

Dear Editor,

Thank you for giving us the opportunity to submit a revised draft of the manuscript “Natural products targeting PI3K/AKT in myocardial ischemic reperfusion injury: A Scoping Review” for publication in the Pharmaceuticals Journal. We appreciate the time and effort that you and the reviewers dedicated for providing feedback on our manuscript and are grateful for the insightful comments on and valuable improvement to our paper. We have incorporated most of the suggestions made by the reviewer, and those changes are highlighted in yellow within the manuscript. Please see below, in blue, for a point-by-point response to the reviewers’ comments and concerns. All line numbers refer to the revised manuscript file.

Reviewer 3

The manuscript entitled: “Natural products targeting PI3K/AKT in myocardial ischemic in reperfusion injury: A Scoping Review” by Syarifah Aisyah Syed Abd Halim et al.  is a scoping review aimed to summarize the studies that evaluated the impact of natural products of the PI3K/AKT pathway in ischemia-reperfusion injury. The Authors evaluated 14 papers, that analyzed 10 natural compounds, by using different methodological approach. In general, this is well written review on an interesting topic. However, in my opinion it should be improved.

Major

  1. The Authors should consider to change the title of their manuscript since the majority of the studies that they analyzed were in vitro Therefore, the term “myocardial” is not appropriate, because it is referred to the whole heart that include different cell types.

We thank reviewer for the comment. Myocardium refers to the muscle layer of the heart and it is made up of cardiomyocytes. Cardiac myoblast is the embryonic precursor of the myocytes.  Also, cardiomyocytes are the contracting cells of the heart and most susceptible to MIRI. The experimental models used in the selected articles are appropriate for the MIRI study. Furthermore, four studies using in vivo method, five studies used in vitro method and another five studies used combined experimental design. We prefer to stick on the current title as the main topic is about the ischemic reperfusion injury in the myocardium.

We hope that reviewer could accept our justification in this matter.

  1. The Authors should stress the limitations of their analysis, underlying the paucity of the available studies in this area, the different experimental models used into the studies, and the heterogeneity of the compounds tested.  

We thank reviewer for the comments. We have added the limitations between lines 463-483.

“The findings from this scoping review are restricted by several factors. First, our search strategy was focused solely on targeting one signaling pathway, namely the PI3K/Akt pathway, while MIRI is a complex injury that involves various other pathways such as TGFβ1/Smads signaling pathway [70], Parkin-mediated mitophagy [71], protein kinase A pathway [72] and activation of endogenous pro survival pathways such as protein kinase G [73]. Therefore, targeting one mechanism at a time may not be sufficient to produce a strong and robust effect in clinical situations where many uncontrolled variables usually coexist. Second, the therapeutic use of natural products was only limited to pre-clinical studies with no clinical trials included. Although all animal models in the papers presented are rodents, there is a difference in the genus and have small sample size, which may limit the generalizability of our findings. Third, several included studies provided experimental values in graphic form and thus the authors were unable to compare the findings in one study to another study that measured the same parameter. Fourth, translation into clinical situations is unfeasible because myocardial infarction is unpredictable, and most of the NPs-based drugs were designed as a pretreatment in MIRI models. Finally, only English articles with accessibility of full text were included. However, given the limitations faced in conducting this scoping review, it is unlikely that any missed data would possibly amend the conclusion drawn based on this review due to a clear focus on pre-clinical studies and the electronic database search was performed to include citations and anywhere the keywords appeared in the article.”

Minor

  1. Since the main topic of this paper is the ischemia/reperfusion injury, and not the myocardial infarction, the first seven lines of the introduction (from line 31 to 37) can be erased.

We thank reviewer for the comment

Myocardial infarction is an irreversible ischemic injury of the myocardium results from the occlusion of coronary artery. Reperfusion approach is one of the treatment given in myocardial infarction. However, this approach can itself induce cardiomyocyte death and known as myocardial ischemic reperfusion injury. In this manuscript, we focus on the natural products that could potentially reduce the complication of MIRI.

We hope that reviewer can accept our justification in this matter.

  1. A brief comment on the beneficial effect of the activation of the PI3K/AKT pathway in ischemia-reperfusion injury should be included in the introduction.

We thank reviewer for the comment. We already add the suggestion between lines 70-72.

“Activation of PI3K/Akt pathway reduced the apoptosis in the cardiomyocytes and mitochondrial oxidative damage leading to better survival of the cardiac function.”

Round 2

Reviewer 1 Report

I am satisfied with the response to my questions and suggestions. 

Reviewer 3 Report

I do not have further comments